# The Prognostic Nutritional Index (PNI): A New Biomarker for Determining Prognosis in Metastatic Castration-Sensitive Prostate Carcinoma

**DOI:** 10.3390/jcm12175434

**Published:** 2023-08-22

**Authors:** Halil Ibrahim Ellez, Merve Keskinkilic, Hüseyin Salih Semiz, Mehmet Emin Arayici, Erdem Kısa, Ilhan Oztop

**Affiliations:** 1Department of Internal Medicine, Division of Medical Oncology, Dokuz Eylül University, Izmir 35340, Türkiye; mervekeskinkilic90@gmail.com; 2Institute of Oncology, Department of Medical Oncology, Dokuz Eylül University, Izmir 35340, Türkiye; hsalihsemiz@hotmail.com (H.S.S.); ilhan.oztop@deu.edu.tr (I.O.); 3Institute of Health Sciences, Department of Preventive Oncology, Dokuz Eylül University, Izmir 35340, Türkiye; mehmet.e.arayici@gmail.com; 4Department of Urology, Tepecik Education and Research Hospital, Health Science University, Izmir 35180, Türkiye; drerdemkisa@hotmail.com

**Keywords:** prostate cancer, prognostic nutritional index, inflammation, prognosis

## Abstract

Prognostic nutritional index (PNI), which is calculated using the albumin level reflecting nutritional status and lymphocyte count reflecting immune status, is useful in showing nutritional and immunological status related to survival and prognosis in many cancers. In this study, we aimed to evaluate the biomarker potential and effect of PNI in determining the prognosis of metastatic castration-sensitive prostate cancer (mCSPC). This retrospective observational study included the complete data of 108 patients with mCPSC who were treated for at least three months between 1 January 2010, and 1 June 2021. The relationships between cancer-specific survival (CSS), overall survival (OS), progression-free survival (PFS), and PNI were evaluated. The Kaplan–Meier method for OS, PFS, and CSS, as well as univariate and multivariate Cox regression models, were used for the statistical analyses. The median age of 108 patients included in the study was 68.54 (61.05–74.19) years. A value of 49.75 was determined to be the best cut-off point for the PNI. OS (months) was found to be significantly lower in patients with low PNI (median: 34.93, 95% CI: 21.52–48.34) than in patients with high PNI (median: 65.60, 95% CI: 39.36–91.83) (*p* = 0.016). Patients with high PNI (median: 48.20, 95% CI: 34.66–61.73) had significantly better CSS (months) than patients with low PNI (median: 27.86, 95% CI: 24.16–31.57) (*p* = 0.001). There was no statistically significant difference in PFS between patients with high PNI values (median: 24.60, 95% CI: 10.15–39.05) and patients with low PNI values (median: 20.03, 95% CI: 11.06–29.03) (*p* = 0.092). The PNI is a good predictor of OS and CSS in patients with mCSPC. The prediction of PFS, albeit showing a trend towards significance, was not statistically significant, probably due to the small number of cases.

## 1. Introduction

Prostate cancer (PCa) is the most common cancer in males. It also ranks second in terms of cancer-related deaths in men [1]. The primary method in the therapy of metastatic castration-sensitive PCa (mCSPC) is testosterone suppressive therapy (androgen deprivation therapy, ADT). ADT can be administered surgically (bilateral orchiectomy) or medically (luteinizing hormone-releasing hormone (LHRH) analogs) [2]. Androgen deprivation therapy (ADT) has been the gold standard of care for patients with mHSPC for almost 80 years. However, while androgen suppression was provided via surgical methods in the past, nowadays, it is provided via medical methods and surgical methods are rarely used. Considering the rather disappointing outcomes of ADT in patients with mHSPC, interest has emerged in the combined use of systemic treatments, including docetaxel or new-generation hormonal agents, e.g., enzalutamide, abiraterone, and apalutamide, to improve survival outcomes in particular [3]. Almost all patients with CSPC progress to metastatic castration-resistant PCa (mCRPC) [4]. In the past, the period of survival once the patients entered the CRPC stage was around 24 months, which was prolonged with the introduction of new-generation hormonal agents into the treatment [5]. Upon passing the metastatic stage, the sites of metastasis and Gleason score are important in determining the disease burden and risk [6]. Although survival has been prolonged with new treatments, the need for a biomarker to predict treatment and castration resistance in the population with metastatic disease is still ongoing [7].

In mCSPC, CHAARTED and LATITUDE study criteria, Gleason score, PSA level, PSA response to treatment, and tumor volume are important criteria for predicting survival and therapeutic benefit. However, the issue of which study criteria are better at predicting survival remains unclear [8]. Recent studies have shown that the prognosis of various cancer types is also affected by patient-related inflammation, immunocompetence, and nutrition. The correlation between nutrition and cancer prognosis is particularly evident [9]. Lymphocyte, neutrophil, thrombocyte, and C-reactive protein levels as nutritional and inflammatory parameters and their use with certain formulas are quite common in cancer patients. Studies on the prognostic value of inflammatory parameters are currently ongoing [10]. Albumin level reflects nutritional status, whereas lymphocyte counts reflect immune status. It is known that albumin levels; lymphocyte count; and their ratios to hematological parameters, such as platelets and neutrophils, have prognostic importance in advanced cancer patients [11]. PNI, which is a marker that can be easily calculated using serum albumin levels and peripheral blood lymphocyte counts, is an important biomarker that has been proven to affect survival in various cancer types [12]. This biomarker is useful in determining the nutritional and immunological status related to survival and prognosis in many cancers [13]. However, there are insufficient clinical studies regarding its association with survival in mCSPC patients. Most studies on this subject have been performed in patients with mCRPC [14].

Although survival in mCSPC has been significantly prolonged with the introduction of new agents into the treatment, there is still a need for novel biomarkers that can predict survival. In this context, PNI, which has been proven to be an effective biomarker for many cancers, comes to the fore. Although there are a number of studies showing that PNI effectively predicts survival in mCRPC patients, there are no studies on its effectiveness in predicting survival in patients with mCSPC, which has a broad clinical spectrum. In light of this information, we hypothesized that PNI could predict the prognosis and survival of mCSPC patients and thus be useful in the follow-up, treatment planning, and management of these patients, and we carried out this study to test this hypothesis by determining the prognostic power of PNI as a novel biomarker in predicting prognosis and survival of mCSPC patients.

## 2. Material and Methods

This retrospective observational study included 201 patients who were diagnosed with prostate adenocarcinoma at the Health Sciences University (HSU) Tepecik Training and Research Hospital between 1 January 2010, and 1 June 2021. The inclusion criteria for the study were as follows: (i) patients diagnosed with stage 4 CSPC, (ii) patients monitored in the clinic for at least three months, (iii) patients who had no therapy for mCSPC (ADT, docetaxel, or any new-generation hormonal agents), and (iv) patients whose PNI score can be calculated from laboratory parameters. At the time of diagnosis, the presence of inflammatory or autoimmune diseases (rheumatoid arthritis, autoimmune hepatitis, diseases that require chronic corticosteroid or immunosuppressive therapy such as rheumatoid arthritis, autoimmune hepatitis, and inflammatory bowel disease), chronic hematological disease, second primary synchronous malignancy (except for carcinoma in situ and non-melanoma skin cancer), systemic therapy for any cancer diagnosis and no remission within the past five years, treatment for serious cardiovascular disease (stage 3 or 4 according to the New York Heart Society classification), and missing data in the hospital database were disqualifying factors for the study. In total, 93 patients who did not meet the specified criteria were excluded from the study and 108 patients who met the study criteria were included. As our study was a retrospective observational study, the sample size was not calculated. The flow diagram of the study is summarized in Figure 1. The study protocol was approved by the HSU Tepecik Training and Research Hospital Non-Interventional Research Ethics Committee, dated 16 August 2021 and numbered 2021/08-07.

The dependent variable in the study was the prognostic nutritional index, calculated using albumin and lymphocyte values at the time of diagnosis. The independent variables were PSA level, systemic therapies added to ADT, age, number of comorbidities, and tumor burden, according to the CHAARTED and LATITUDE trial criteria. Sociodemographic and clinicopathological data as well as laboratory parameters of the patients were obtained retrospectively from the hospital database. The current laboratory parameters at diagnosis (before prostate biopsy) of the 108 patients who met the inclusion criteria were recorded. The PNI was calculated as 10 × serum albumin (g/dL) + 0.005 × total lymphocyte count (per mm^3^) [13]. To evaluate the presence of clinical metastases, contrast-enhanced computed tomography of the abdomen and thorax, bone scintigraphy, and PSMA/PET results in the hospital database were performed. The patients were grouped into low- and high-risk groups according to the CHAARTED trial criteria and into low- and high-risk groups according to the LATITUDE trial criteria.

CSS data were obtained by calculating the time from the date of initiation of systemic therapies for metastatic disease to the date of death or the last patient visit, whichever occurred first. OS data were obtained by calculating the time from the first diagnosis to death or last follow-up visit. The time elapsed from the date of therapy initiation with the diagnosis of CSPC to the first PSA elevation or the first radiological progression (whichever occurred first) was considered progression-free survival (PFS).

### Statistical Analysis

A receiver operating characteristic (ROC) curve analysis was applied to select the most appropriate cut-off point for PNI to identify patients at a high risk of cancer-related death. Accordingly, the best cut-off point for PNI was determined as “49.75”. The patients were divided into two groups, those with a value at or above 49.75 and those below the 49.75 value, and then compared in terms of OS, CSS, and PFS. OS (overall survival) refers to the time from disease diagnosis to death. CSS was defined as the time from metastasis development to death, and statistical computations were performed based on this timeframe. The time from the diagnosis of CSPC to the start of treatment until the occurrence of the first PSA increase or first radiological progression (whichever occurred first) was considered PFS (or DFS). In addition to descriptive statistics, the chi-square and Fisher’s exact tests were used for categorical variables to evaluate the data. The conformity of continuous data to normal distribution was quantified using the Kolmogorov–Smirnov and Shapiro–Wilk tests. The Mann–Whitney U test was used to determine differences between the variables indicated based on the measurements. The Kaplan–Meier method was used to estimate CSS, PFS, and OS, while the log-rank test was used to investigate differences in survival. To evaluate the effect of the PNI on survival, univariate and multivariate Cox regression models were used to identify the best predictive variables. The Median follow-up time in the study was calculated using reverse Kaplan–Meier. SPSS (version 25.0) was used to analyze all data. Statistical significance was determined as *p* < 0.05 in all tests.

## 3. Results

The median age of the 108 patients included in this study was 68.54 (61.05–74.19) years. The median PSA at the time of diagnosis was 110.60 (35.28–154.52) μg/L. The rate of patients who received chemotherapy during the castration-sensitive stage was 44.4% (n = 48). Of the 108 patients, 93 (86.1%) had no history of primary surgery. Additionally, 50 (46.3%) patients did not have any comorbidities, 36 (33.3%) had one comorbidity and 22 (20.4%) had two or more comorbidities. The sociodemographic and clinicopathological characteristics of the patients are shown in Table 1.

The optimal PNI cut-off value for CSS was determined as 49.75 based on the ROC curve analysis. There was no significant difference between the PNI groups in terms of serum PSA level at diagnosis; LATITUDE study risk levels, i.e., low- or high-risk; or age (*p* > 0.05) (Table 2). On the other hand, in terms of CHAARTED study risk levels, patients with high (>49.75) PNI values had low-volume disease, whereas patients with low (<49.75) PNI values had high-volume disease, indicating a significant difference between the PNI groups (*p* = 0.002). In addition, the analysis of metastatic sites (M1a, M1b, and M1c) in terms of PNI values revealed that PNI values decreased as the metastatic site progressed from M1a to M1c (*p* = 0.017). A comparison of the groupings made according to the LATITUDE and CHAARTED criteria showed that 27.3% of the patients identified as low-risk patients according to the LATITUDE criteria had high-volume disease according to the CHAARTED criteria and 3.2% of the patients identified as high-risk patients according to the LATITUDE criteria had low-volume disease according to the CHAARTED criteria; thus, there was a significant incompatibility between the two systems (*p* < 0.001).

OS (months) was significantly lower in patients with low PNI values (median: 34.93, 95% confidence interval (CI): 21.52–48.34) compared with patients with high PNI values (median: 65.60, 95% CI: 39.36–91.83) (*p* = 0.016) (Figure 2). Patients with high PNI values (median: 48.20, 95% CI: 34.66–61.73) had significantly better CSS (months) than those with low PNI values (median: 27.86, 95% CI: 24.16–31.57) (*p* = 0.001) (Figure 3). On the contrary, there was no statistically significant difference in PFS between patients with high PNI values (median: 24.60, 95% CI: 10.15–39.05) and patients with low PNI values (median: 20.03, 95% CI: 11.06–29.03) (*p* = 0.092) (Figure 4). Although we observed a trend (patients with high PNI values showed higher PFS than those with low values), the data did not reach statistical significance, probably because of the small sample size.

OS in the high and low PNI groups was analyzed according to metastasis sites (M1a, M1b, and M1c). Longer OS was observed in patients with high PNI values (Figure 5) than in patients with low PNI values (Figure 6) in each metastasis site group (*p* < 0.001). Additionally, an analysis of the OS in the high- and low-PNI groups according to CHAARTED criteria revealed significantly higher OS in patients with low-volume disease than in patients with high-volume disease (*p* < 0.001) (Figure 7 and Figure 8). Similarly, an analysis of the OS in the high- and low-PNI groups according to LATITUDE criteria revealed significantly higher OS in low-risk patients than in high-risk patients (*p* < 0.001) (Figure 9 and Figure 10).

The univariate analyses revealed significant correlations between decreased PNI, high-volume disease, high-risk statuses, poorer OS (*p* = 0.018, *p* < 0.001, and *p* < 0.001, respectively), and poorer CSS (*p* = 0.001, *p* < 0.001, and *p* < 0.001, respectively) (Table 3). The International Society of Urological Pathology (ISUP) grades 4 and 5 were associated with worse prognosis only for OS (*p* = 0.006). In the multivariate analyses, the hazard ratios (HRs) of PNI were 2.280 (95% CI: 1.285–4.046) for OS and 3.011 (95% CI: 1.664–5.447) for CSS (Table 4). Age was determined to be an independent prognostic factor for OS and CSS (HR: 1.040; 95% CI: 1.001–1.080, *p* = 0.042, and HR:1.040, 95% CI: 1.003–1.078, *p* = 0.034, respectively).

## 4. Discussion

Owing to the new treatment modalities developed and widely used in recent years, the median OS of PCa patients has exceeded five years [15]. Many studies have shown that chemo-hormonal therapy with docetaxel and ADT, which is widely applied especially in the mCSPC stage, prolongs OS for approximately 17 months [16]. Similar results were obtained with treatments involving abiraterone, enzalutamide, and apalutamide in patients with mCSPC [17]. Many biomarkers have been studied to determine the patients who would benefit more from these therapies and would live longer. However, to date, no biomarker has been found that can replace the parameters currently used to determine prognosis, namely the CHAARTED and LATITUDE criteria, Gleason score, PSA level, PSA response to therapies, and TNM stage. Therefore, the need for a novel biomarker to predict treatment and castration resistance in the patient population with metastatic disease remains. In this context, we aimed to evaluate the potential of PNI as a novel biomarker in predicting the prognosis in mCSPC patients and found that PNI affects survival in patients diagnosed with mCSPC, independent of all other prognostic factors. Various optimal PNI cut-off values were used in studies investigating PNI in the context of different cancer types in the literature. In one of these studies conducted by Li et al. (2020) with 208 patients, the optimal PNI cut-off value was determined as 50.2. Accordingly, they reported that patients with PNI values equal to or above 50.2 had better CSS, OS, and PFS than patients with PNI values below 50.2 [18]. They also found a significant difference between patients with high and low PNI values in OS and CSS, regardless of M1a, M1b, and M1c mutations. In comparison, the optimal PNI cut-off value used in this study was determined as 49.75 based on the ROC curve analysis. In parallel with the said study, the CSS and OS of patients with PNI values equal to or above 49.75 were significantly better than those with PNI values below 49.75. On the other hand, there was also a positive correlation with PNI and PFS; however, unlike said study, it did not reach statistical significance possibly due to the relatively small sample size.

Recent data have suggested that systemic inflammatory response plays an important role in the development and progression of cancer. It has been hypothesized that tumors benefit from inflammatory processes in their microenvironment [19]. These processes provide bioactive molecules, including growth factors, survival factors, proangiogenic factors, and extracellular matrix enzymes, which promote invasion, angiogenesis, and metastasis. The PNI, which was initially introduced by Onodera et al. [12] to assess the immunological and nutritional status of patients undergoing gastrointestinal surgery, is a biomarker that combines albumin level and lymphocyte count. Mohri et al. reported that PNI is an independent predictor of postoperative severe complications and has a prognostic value comparable with that of carcinoembryonic antigen (CEA) level and postoperative TNM staging [20].

It is now known that the lymphocyte count and the lymphocyte count around the tumor (tumor-infiltrating lymphocytes) have prognostic importance [21]. The blood albumin level has also been reported to have prognostic importance. Low lymphocyte counts in the blood may indicate that the antitumor immune response against the tumor will be weak [22]. In parallel, it is known that the inflammation caused by the tumor and the cytokines released secondary to this inflammation reduce blood albumin levels and may thus play a role in tumor progression [23]. An in vitro study on prostate cancer cell lines (from androgen-insensitive as well as androgen-sensitive tumors) showed that a reduction in prostate inflammation using herbal extracts is linked to tumor apoptosis and death through a specific mitochondrial pathway [24]. In light of the foregoing findings, PNI, as a biomarker that combines the prognostic power of both albumin level and lymphocyte count, is likely to have a high potential in predicting survival independently of all factors.

Using the mean or cut-off values of PNI reported in studies available in the literature for other cancer types may not be useful in assessing the true prognostic value of PNI. The prognostic value of a new biomarker should first be evaluated as a continuous variable. Using this approach, we showed in both univariate and multivariate analyses that lower PNI values were significantly associated with poorer DFS and CSS. As clinical decision-making is frequently based on cut-off values, we evaluated the optimal PNI cut-off value for prognostic stratification using a discrimination analysis. Elucidating the relationship between a potential new biomarker and prognosis alone is not sufficient for clinical decision-making.

The prognostic and predictive importance of the criteria developed in the LATITUDE and CHAARTED studies have been established. However, there are significant discrepancies between the predictions made using said criteria [25]. As a matter of fact, a comparison of the groupings made according to the LATITUDE and CHAARTED criteria showed that 27.3% of the patients identified as low-risk patients according to the LATITUDE criteria had high-volume disease according to the CHAARTED criteria and 3.2% of the patients identified as high-risk patients according to the LATITUDE criteria had low-volume disease according to the CHAARTED criteria, indicating the significant incompatibility between these two systems. Additionally, no significant relationship was found between the grouping made according to the LATITUDE criteria and PNI values. On the other hand, there was a significant relationship between the grouping made according to the CHAARTED criteria and PNI values. Accordingly, there was a significantly higher number of low-volume disease patients than high-volume disease patients in the high-PNI (≥49.75) group than in the low-PNI (<49.75) group. Contrary to these findings, the univariate analysis revealed significant relationships between survival and the grouping made according to both the LATITUDE and CHAARTED criteria. This finding may be attributed to the discrepancy between the LATITUDE and CHAARTED criteria or an insufficient number of patients. These two prognostic criteria should be further investigated in large-scale studies.

Another important finding of our study is that only 58.3% of the patients who progressed to the mCRPC stage were able to receive first-line therapy and 20.4% received second-line systemic therapy for mCRPC. In other words, when a patient enters the castration resistance stage, the likelihood of receiving systemic therapy as well as survival time is reduced, increasing the value of PNI as a potential biomarker [26].

There were some limitations to this study. The primary limitation of this study was its retrospective design, rendering its reliability lower than that of a prospective study. Secondly, the single-center design, relatively short follow-up period, and small sample size may be deemed additional limitations of this study. Thirdly, most patients included in this study were diagnosed before the CHAARTED and LATITUDE studies were carried out. Hence, the fact that systemic therapies were not added to ADT in most patients in the castration-sensitive stage might have affected the results. Fourthly, many factors such as nutritional status, infection, and inflammation might have affected PNI. Given its retrospective design, the findings of this study did not shed light on whether providing nutritional support to patients with low PNI values could improve PNI or survival.

## 5. Conclusions

The first point to state is that the PNI is low-cost, is easy to calculate, adds straightforward clinical parameters, and is easily available in low-resource centers as well as in simplified clinical settings. Its potential role as a clinical prognostic biomarker in patients with mCSPC deserves further investigation with longer follow-up and larger patient groups. It is the authors’ belief that more studies are needed in the future in this interesting field to obtain more accurate and clinically relevant data and to improve the prognostic characterization of the population of patients with metastatic prostate cancer, especially when undergoing novel treatment strategies.

## Figures and Tables

**Figure 1 jcm-12-05434-f001:**
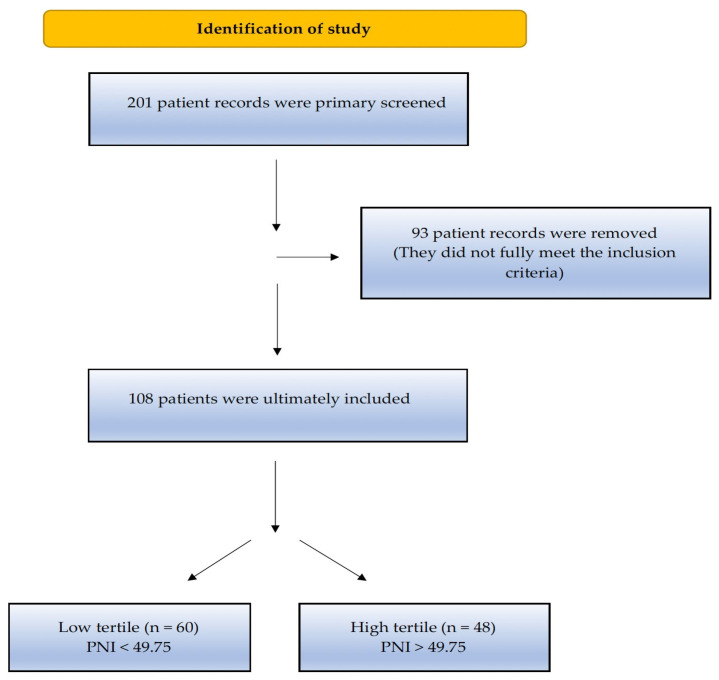
Flow chart of the study according to the CONSORT diagram.

**Figure 2 jcm-12-05434-f002:**
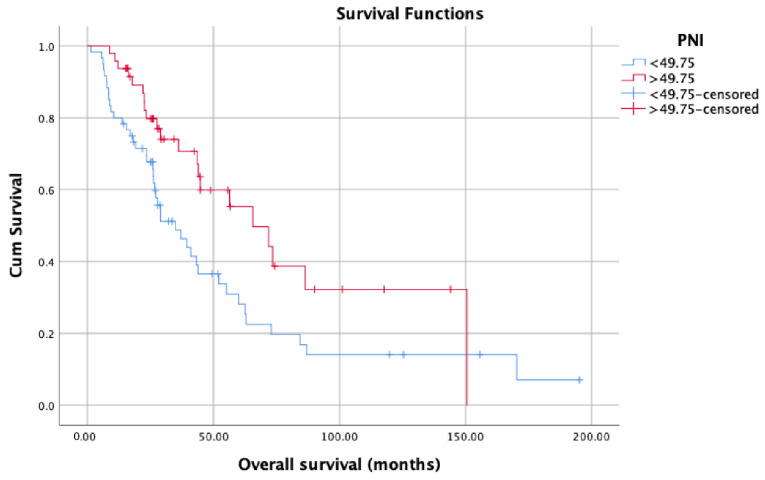
Overall survival in patients with low PNI compared with patients with high PNI.

**Figure 3 jcm-12-05434-f003:**
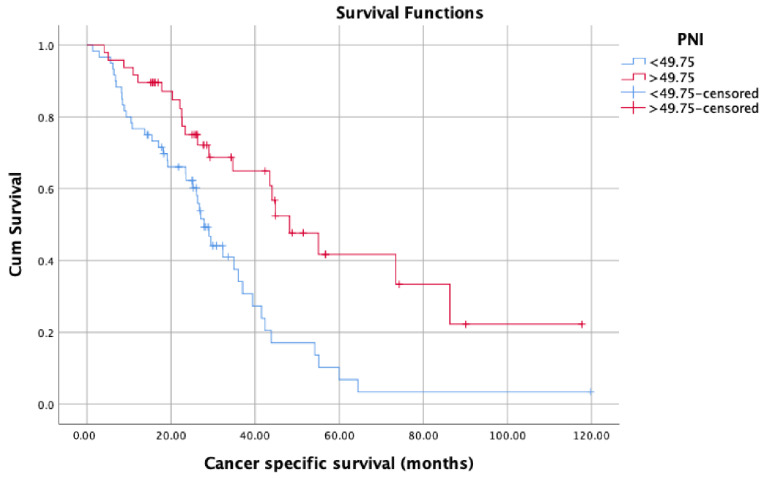
Cancer-specific survival in patients with low PNI compared with patients with high PNI.

**Figure 4 jcm-12-05434-f004:**
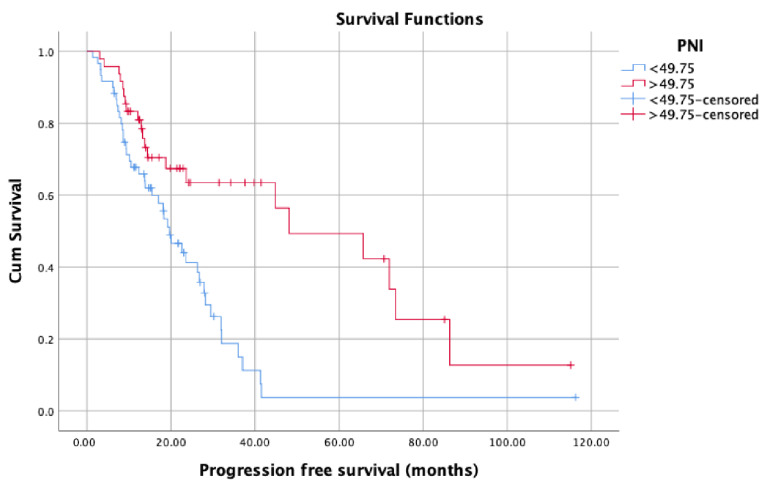
Progression-free survival in patients with low PNI compared with patients with high PNI.

**Figure 5 jcm-12-05434-f005:**
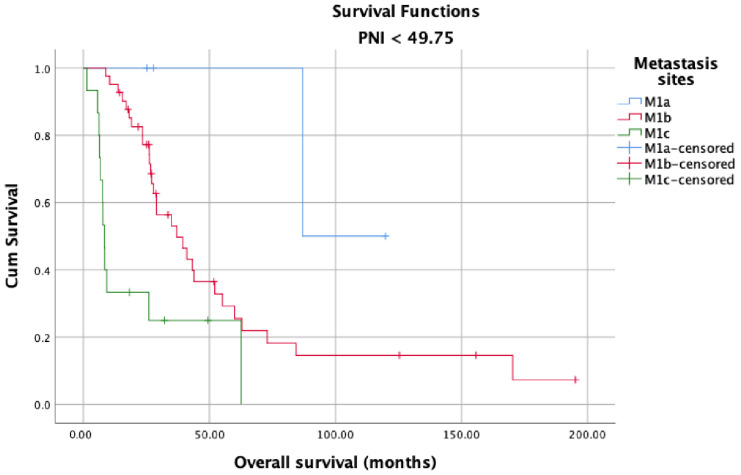
Overall survival in the low PNI groups according to metastasis site (M1a, M1b, and M1c).

**Figure 6 jcm-12-05434-f006:**
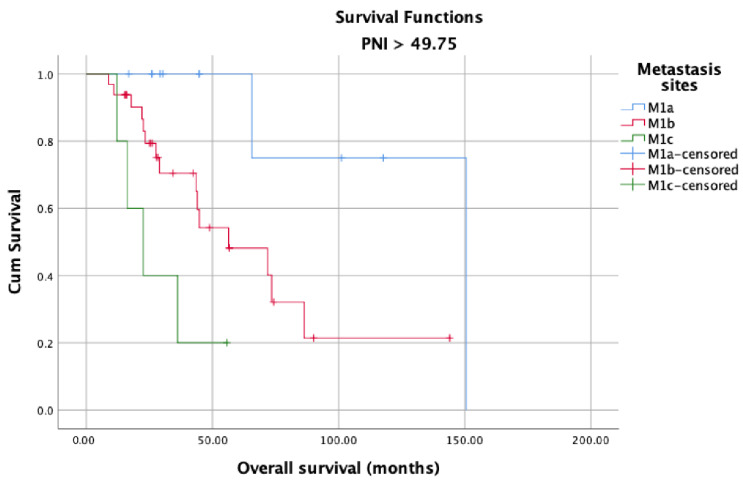
Overall survival in the high PNI groups according to metastasis site (M1a, M1b, and M1c).

**Figure 7 jcm-12-05434-f007:**
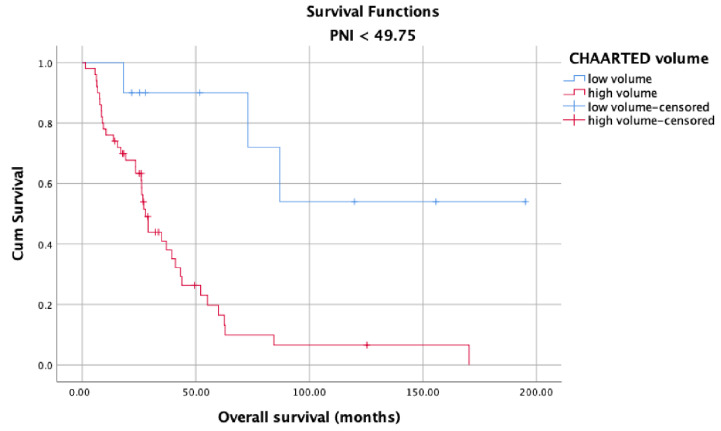
Overall survival in the low PNI group according to CHAARTED volume levels.

**Figure 8 jcm-12-05434-f008:**
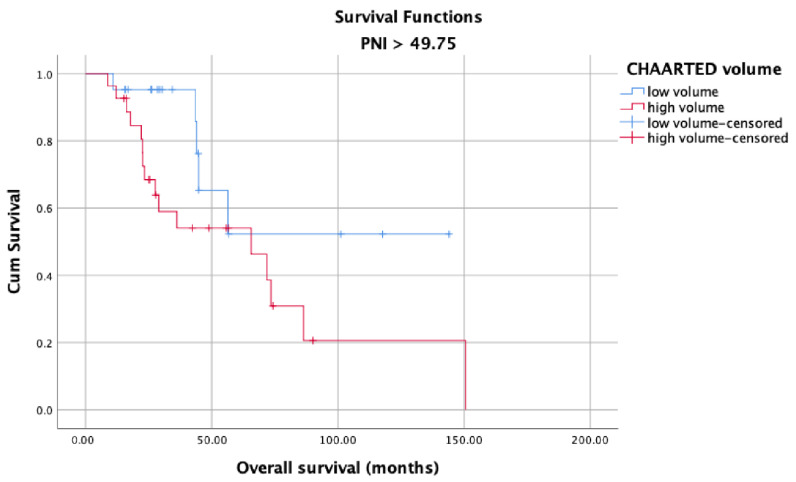
Overall survival in the high PNI group according to CHAARTED volume levels.

**Figure 9 jcm-12-05434-f009:**
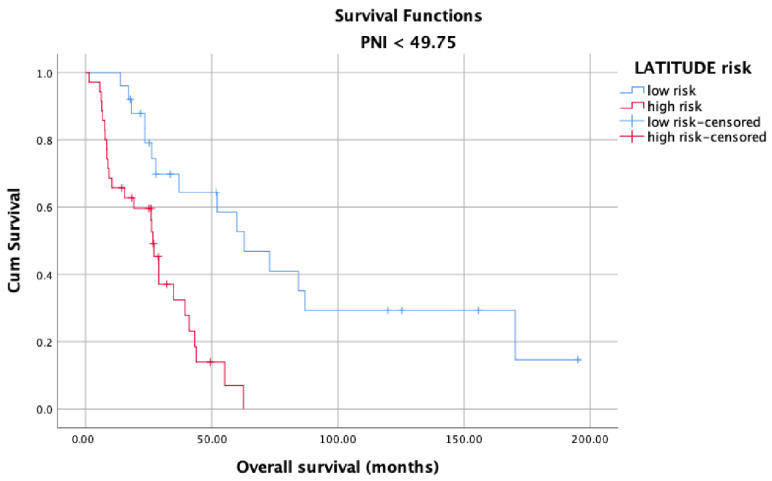
Overall survival in the low PNI group according to LATITUDE risk levels.

**Figure 10 jcm-12-05434-f010:**
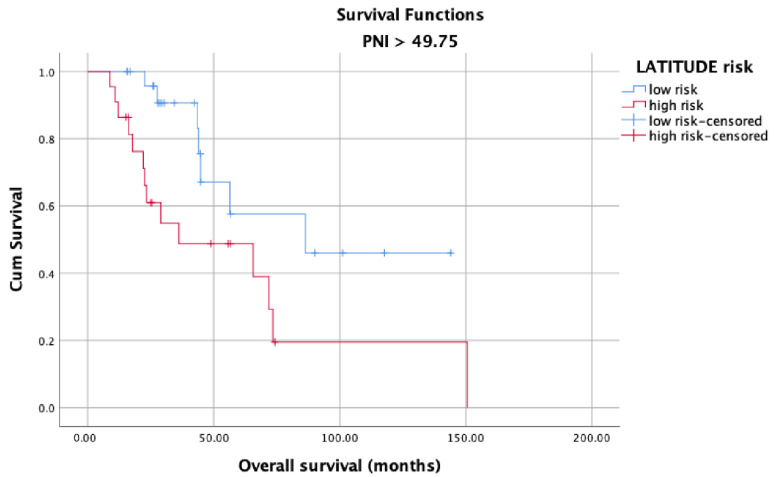
Overall survival in the high PNI group according to the LATITUDE risk levels.

**Table 1 jcm-12-05434-t001:** Sociodemographic and clinicopathological characteristics of the patients.

Variables	Total (n = 108)
Age group, n (%)	
<70	63 (58.3%)
≥70	45 (41.7%)
Survival Status, n (%)	
Alive	45 (41.7%)
Ex	63 (58.3%)
Received chemotherapy in the castration-sensitive stage, n (%)	
Yes	48 (44.4%)
No	60 (55.6%)
Disease severity grade (ISUP) groups, n (%)	
1	1 (0.9%)
2	12 (11.1%)
3	16 (14.8%)
4	24 (22.2%)
5	49 (45.4%)
History of primary surgery, n (%)	
None	93 (86.1%)
Radical	15 (13.9%)
CHAARTED criteria groups, n (%)	
Low-volume disease	31 (28.7%)
High-volume disease	77 (71.3%)
LATITUDE criteria groups, n (%)	
Low risk	51 (47.2%)
High risk	57 (52.8%)
PNI groups, n (%)	
<49.75	60 (55.6%)
>49.75	48 (44.4%)

**Table 2 jcm-12-05434-t002:** Clinical characteristics of patients with prostate cancer according to the PNI.

	PNI Groups				*p* Value *
	≥49.75		<49.75		1.000
Age groups	n	%	n	%	
<70	28	44.4	35	55.6	
≥70	20	44.4	25	55.6	
	PNI Groups				*p* value **
	≥49.75		<49.75		0.127
PSA (median, percentiles (25, 75), μg/L)	99.42 (20.49–156.25)		126.19 (56.70–154.52)		
	PNI Groups				*p* value *
	≥49.75		<49.75		0.002
CHAARTED criteria	n	%	n	%	
Low volume	21	67.7	10	32.3	
High volume	27	35.1	50	64.9	
	PNI Groups				*p* value *
	≥49.75		<49.75		0.196
LATITUDE criteria	n	%	n	%	
Low risk	26	51.0	25	49.0	
High risk	22	38.6	35	61.4	
	PNI Groups				*p* value *
	≥49.75		<49.75		0.017
Metastasis group	n	%	n	%	
M1a	11	73.3	4	26.7	
M1b	32	43.8	41	56.2	
M1c	5	25.0	15	75.0	
	LATITUDE risk				*p* value *
	Low risk		High risk		<0.001
CHAARTED volume	n	%	n	%	
Low volume	30	96.8	1	3.2	
High volume	21	27.3	56	72.7	

Abbreviations: PNI: prognostic nutritional index; PSA: prostate-specific antigen; ISUP: International Society of Urological Pathology; CHAARTED: Chemo-hormonal Therapy versus Androgen Ablation Randomized Trial for Extensive Disease in Prostate Cancer; LATITUDE: Long-Acting Therapy to Improve Treatment Success in Daily Life; M1a: the cancer has spread to lymph nodes away from the groin area; M1b: the cancer has spread to the bones, M1c: the cancer has spread to another part of the body, with or without spread to the bones. * Pearson’s chi-square test. ** Mann–Whitney U test.

**Table 3 jcm-12-05434-t003:** Univariate analysis of various clinical parameters in patients with prostate cancer.

Parameter	Overall Survival (OS)		Cancer Specific Survival (CSS)	
	HR (95% CI)	*p* Value	HR (95% CI)	*p* Value
Age (years)	1.020 (0.988–1.053)	0.223	1.019 (0.988–1.050)	0.226
PSA (μg/L)	1.000 (1.000–1.001)	0.555	1.001 (1.00–1.001)	0.847
PNI		0.018		0.001
>49.75	1		1	
<49.75	1.893 (1.117–3.208)		2.460 (1.440–4.202)	
ISUP grade group		0.006		0.102
1–3	1		1	
4–5	2.467 (1.304–4.668)		1.648 (0.906–2.998)	
CHAARTED		<0.001		<0.001
Low volume	1		1	
High volume	4.249 (2.013–8.965)		3.980 (1.831–8.348)	
LATITUDE		<0.001		<0.001
Low risk	1		1	
High risk	3.322 (1.926–5.731)		2.921 (1.718–4.966)	

Abbreviations: OS: overall survival, CSS: cancer-specific survival, HR: hazards ratio, CI: confidence interval, PSA: prostate-specific antigen, PNI: prognostic nutritional index, ISUP: International Society of Urological Pathology, CHAARTED: Chemo-hormonal Therapy versus Androgen Ablation Randomized Trial for Extensive Disease in Prostate Cancer, LATITUDE: Long-Acting Therapy to Improve Treatment Success in Daily Life.

**Table 4 jcm-12-05434-t004:** Multivariate analysis of various clinical parameters in patients with prostate cancer.

Parameter	Overall Survival (OS)		Cancer Specific Survival (CSS)	
	HR (95% CI)	*p* Value	HR (95% CI)	*p* Value
Age (years)	1.040 (1.001–1.080)	0.042	1.040 (1.003–1.078)	0.034
PSA (μg/L)	1.000 (1.000–1.001)	0.781	1.001 (1.00–1.001)	0.835
PNI		0.005		<0.001
>48.9	1		1	
<48.9	2.280 (1.285–4.046)		3.011 (1.664–5.447)	
ISUP grade group		0.001		0.025
1–3	1		1	
4–5	2.863 (1.501–5.459)		2.014 (1.093–3.711)	

Abbreviations: OS: overall survival, CSS: cancer-specific survival, HR: hazards ratio, CI: confidence interval, PSA: prostate-specific antigen, PNI: prognostic nutritional index, ISUP: International Society of Urological Pathology.

## Data Availability

The datasets used and/or analyzed during the current study are available from the corresponding author upon reasonable request.

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
