# Peer review of "The Prognostic Nutritional Index (PNI): A New Biomarker for Determining Prognosis in Metastatic Castration-Sensitive Prostate Carcinoma"

_jcm, 2023, doi:10.3390/jcm12175434_

Round 1

Reviewer 1 Report

The Authors provide an interesting study on the prognostic role of PNI in  clinical management of men who have been diagnosed with mCSPC, reporting the data on 108 consecutive patients treated at a single institution over 2 consecutive years (2019-2021), for at  least three consecutive months. This is the starting point. The paper is potentially very interesting and clinically significant. However, there are a number of points which must be clarified and better conceptualized, before the paper can be considered for publication in JCM.

These points are detailed below for the Authors:

ABSTRACT. Suggested changes:

Background: The first line is not correctly stated.  The Background is not the summary of the paper, but it should contain the relevant synthetic knowledge currently available about the subject (like, for example, what is currently known about PNI, what is missing, and then what is the aim of this study).

Conclusions: PNI is a strong predictor of OS and CSS in patients with mCSPC. Prediction of PFS, albeit showing a trend towards significance, did not result statistically significant, probably due to the small number of cases.

INTRODUCTION:.

Could be added to the Introduction: Prostate cancer is the most common malignancy in men. Despite considerable progress in its management, predicting the outcome is still difficult for individual patients, given the variable clinical history of the disease. Nutritional and environmental factors can contribute substantially to the clinical course of prostate cancer.

Line 35-39: re-phrase more accurately and do not simplify the clinical problem. First state the protocols used with metastatic prostate cancer patients in current clinical practice (bilateral orchiectomy is seldomly used, and cannot be compared to medical therapy). Then do not forget to underline that, despite the introduction of additional therapies, we still lack biomarkers able to predict the outcome in the metastatic population. This concept, and the relevant citations, must be added by the Authors.

Line 84-86: re-phrase, unclear. Do not use the generic expression “does not have significant biomarkers”, but clarify this concept more specifically.

At end of the Introduction, explain why your own study was started, and then add that it shows a novel information, which the PNI potential significance as a biomarker. If this is quite original data, not reported in previous studies, just underline it. Explain which gap is your study going to fill (this is one of the strengths of your own study). Elsewhere in the paper, like in the Conclusion or Discussion, you could recall this concept and expand it, citing the relevant references.

RESULTS

Line 167-168: Re-phrase, unclear

Line 179: Do not start the sentence with “In addition”, but rather with “On the contrary”.

Line 182-183: Although we observed a trend (patients with high PNI values showed higher PFS as compared to those with low values), the data did not reach statistical significance, probably do to the small sample size.

DISCUSSION: 

General comment: The discussion section is the place where your results are compared to what has been published before. Here you must stress the strength and the originality of your own work, confronting it with updated references and recent citations.

Line 229-231:re-phrase better, unclear. Specify better the clinical concept you want to emphasise.

Line 240-242: Re-phrase more accurately. What do you mean by “that can simply be measured in blood”?.

Line 245-246: Add here the citation about this statement

Line 247-249: It is now known that inflammation caused by the tumor and inflammation-related citokines not only reduce blood albumin levels, but can eventually play a role in tumor progression. It has been shown in an in vitro study on prostate cancer cell lines (from androgen-insensitive as well as androgen-sensitive tumors) that reduction of prostate inflammation using herbal extracts is linked to tumor apoptosis and demise through a specific mitochondrial pathway (add citation here: Baron A., et al, BJU Int 103:1275-1283;2009)

Line 261-264: Re-phrase, unclear

Line 267-269: What do you mean exactly? Explain this concept more clearly.

Line 274-278: Limitations of the study, monocentric, small study group, short follow up. At the end, add the other limitations you mention before.

CONCLUSIONS

The first point to state is the PNI is a low-cost, easy to calculate add straightforward clinical parameter, available easily in low-resources centres as well as in simplified clinical settings. Its potential role as a clinical prognostic biomarker in patients with mCSPC deserves further investigation, with longer follow-up and larger patients’ group. It is the Authors’ belief that more studies are needed in the next future in this interesting field, to obtain more accurate and clinically relevant data and improve the prognostic characterisation of the population of patients with metastatic prostate cancer,  especially when undergoing novel treatments strategies.

The English language in the paper needs moderate revision. The points more in need of language revision and improvement have been shown to the Authors in the previous paragraph

Reviewer 2 Report

The methods are generally appropriate, this article has several flaws, specific comments follow,

Abstract

Please follow the guideline to change the abstract.

Introduction

Please note the scientific writing (or scientific editing) for each sentence and cite references.

Materials and Methods

Please add the flowchart more concisely through this study.

Please add that the relationship between  Prognostic Nutritional Index (PNI) and disease-free survival (DFS) was analyzed by statistical analysis that’s more robustness.

Please add the study protocol which was approved by the institutional committee on human research of the Institutional Review Board (IRB).

Results

Please modify the table more concisely through this study.

Discussion

How to apply "Prognostic Biomarker" to characterize or distinct surrogate in clinical application through this study.

Molecular-biological probes are important in cancer detection, please discuss  molecular-biological processes and  show the  example of histological and morphological characteristics in the metastatic castration-sensitive prostate carcinoma tissues.

The pre-operative PNI can better reflect the surgical risk and nutritional status of cancer patients, please inspect the detail as follows:

*Correlations between the PNI and clinical characteristics such as pre-operative PNI level were associated with tumor node, and metastasis stages, tumor differentiation, patients age and lymph node metastasis.

*Characterizing the relationship of biomarkers to clinical outcomes such as histological and morphological characteristics rationale.

*Relationships between PNI and clinicopathological features in patients or check/cite the relative references with this article.

Please follow the guideline to change the abstract and note the scientific writing for each sentence, paragraph and cite reference. 

Round 2

Reviewer 2 Report

(1) Abstract shouldn't add the background (Line 13).

(2) All of the table presentation is no good in this article, please modify or check others literature in scientific editing. The author does not use a uniform format in this article, such as P-values, p-values ​​(upper and lower case) in tables and texts, and some letters require spaces in sentences. Table 1. Patients’ sociodemographic and clinicopathological characteristics please follow the academic scientific writing to modify symbols.

(3) This study was a retrospective study, and ethical approval has been obtained. Please show your approval number in this cohort selection.

(4) Please double check all details for each sentences, paragraphs and references before submission to avoid invalid revisions.

Please follow the academic scientific writing.

Author Response

(1) Abstract shouldn't add the background (Line 13)

Response to comment:

the summary has been edited as you requested.

(2) All of the table presentation is no good in this article, please modify or check others literature in scientific editing. The author does not use a uniform format in this article, such as P-values, p-values ​​(upper and lower case) in tables and texts, and some letters require spaces in sentences. Table 1. Patients’ sociodemographic and clinicopathological characteristics please follow the academic scientific writing to modify symbols.

Response to comment:

Thank you for your valuable suggestions. All tables have been meticulously reworked in line with the suggestions. The problems in our article have been reorganized according to academic writing rules.

(3) This study was a retrospective study, and ethical approval has been obtained. Please show your approval number in this cohort selection.

Response to comment

Ethics committee approval is attached. The approval number is also indicated in the article.

(4) Please double check all details for each sentences, paragraphs and references before submission to avoid invalid revisions.

Response to comment:

The whole article has been reworked. Corrections were made to the entire structure. References were reviewed at least 3 times and reorganized. Thank you for your valuable suggestion.

Our article was a much better quality job with your valuable suggestions. Thank you for devoting your valuable time to us

Best regards

Halil Ibrahim Ellez, MD
